# Impact of the Temperature in Endophytic Ascomycota Isolated from Antarctic Hair-Grass

**DOI:** 10.3390/life12101501

**Published:** 2022-09-27

**Authors:** Fabíola Lucini, Guilherme Afonso Kessler de Andrade, Filipe de Carvalho Victoria, Margéli Pereira de Albuquerque

**Affiliations:** 1Núcleo de Estudos da Vegetação Antártica—NEVA, Federal University of Pampa (UNIPAMPA), Street Aluízio Barros Macedo Br 290, São Gabriel 97300-000, Brazil; 2Programa Antártico Brasileiro—PROANTAR, Esplanada dos Ministérios, Brasília 70055-900, Brazil

**Keywords:** endophitic fungi Phylogeny, biodiversity, antarctic ecology, molecular biology, ITS, nLSU

## Abstract

Antarctica is one of the most inhospitable continents on the planet, with lichens and mosses being the most common terrestrial organisms in ice-free areas. Antarctica is represented by only two species of Angiosperms, *Deschampsia antarctica* Desv. (Poaceae) and *Colobanthus quitensis* (Kunth) Bartl. (Caryophyllaceae). In this study, we characterized fungi isolated from the fresh leaves of this grass species. The fungi were isolated from four individual plants from Half Moon Island (246 leaf fragments investigated), and seven from King George Island-Keller Peninsula (with 111 leaf fragments investigated) Antarctica. *Neoascochyta paspali*, *Septoriella elongata*, *Pyrenophora* cf. *chaetomioides* and *Alternaria* sp. were associated with the plant and identified through analysis of the sequences of the internal transcribed spacer region (ITS) of the rDNA and nuclear ribosomal large subunit rRNA gene (LSU) as well as through macro and micro-morphological characteristics. The isolates showed higher growth rate ranging from 10 to 20 °C. An interesting result was that the aforementioned fungi are already recognized as both plant pathogens and endophytic fungi. The results demonstrate that *D. antarctica* is an interesting fungal source. Those species might provide important information about the relationship on the endemic Antarctic biota.

## 1. Introduction

The Antarctic region plays a key role in the balance of atmospheric and climatic dynamics [1]. However, the Antarctic continent is considered one of the most inhospitable ecosystems, being the coldest, windiest, and driest, with a high incidence of radiation, all of which restrict the development of many life forms [2].

The Antarctic terrestrial diversity is predominantly composed of lichens and bryophytes (mosses and liverworts) species and includes only two species of native vascular plants: *Deschampsia antarctica* Desv. (the Antarctic hair-grass—Poaceae) and *Colobanthus quitensis* (Kunth) Bartl. (Caryophyllaceae) [3,4]. *Deschampsia antarctica* is a pioneer species that colonises exposed environments after retraction of the glaciers. It can grow over dead mosses or directly on live moss carpets [5,6,7].

Several studies to date have demonstrated the ability of this Antarctic hairgrass to survive extreme conditions or the mechanisms involved in this resistance [8,9,10]. However, few studies have investigated its interaction with the associated fungi. Plant pathogens within the Antarctic region as well known and are mostly identified in lichens and mosses, more so than in angiosperms [4]. For other substrates, fungi have been reported from soil, woody components, rocks and macroalgae [11].

For *D. antarctica*, associated fungi have been reported in some studies, indicating a wide range of mycorrhizae [12] as well as endophytes, parasites, and predatory fungi [13,14,15,16].

The fungi associated with Antarctica plants typically include yeasts and filamentous fungi consisting of species from Chytridiomycota, zygomycetous fungi, Glomeromycota, Basidiomycota, and Ascomycota [17]. Ascomycota represents the major group of endophytic fungi isolated to date [18]. The association can be endophytic, such as with pathogens or decomposers [19,20].

In this study, were evaluated the growth of four endophitic fungi associated with *D. antarctica*, collected in the Antarctic summer of 2014 and 2016 from Half Moon and King George Islands. The isolated fungi were identified by molecular taxonomy and by macro- and micro-morphological characterization and represented the genus *Pyrenophora*, *Phaeosphaeria*, *Neoascochyta*, and *Alternaria*. In addition, mycelial growth was assessed at four different temperatures and in different culture media.

## 2. Materials and Methods

### 2.1. Study Area

The South Shetlands Archipelago is in the Maritime Antarctic, lying in the Northwest of the Antarctic Peninsula. It is composed of 8 large islands and many other smaller ones. The small Half Moon Island is located at 62°36′ S; 59°53′ W and King George Island at 62°01′21″ S, 58°15′05″ W [21], where specimens of *D. antarctica* were collected.

### 2.2. Plant Material and Isolation of the Fungi

Fresh *D. antarctica* leaves were collected from plants growing under natural conditions in Half Moon Island during the antarctic summer of 2014, and from King George Island (Keller Peninsula) during the antarctic Summer of 2016. The plant material was stored in sterile plastic bags, frozen, and transported to Brazil. The leaves were sterilized by successive immersion in 70% ethanol (1 min) and 2% sodium hypochlorite (3 min), followed by a sterile distilled water rinse (2 min) [14]. The fragments were then plated on Petri dishes containing Potato Dextrose Agar (PDA, Merck^®^ KGaA, Darmstadt, Germany) supplemented with chloramphenicol (100 µg mL^−1^). The plates were incubated up to 60 days at 20 °C, and individual colonies were transferred to PDA, and stored at 20 °C. The long-term preservation of mycelial material was performed using the Castellani and Mineral Oil methods [22]. The fungi were isolated from four individual plants in Half Moon Island (246 leaf fragments investigated), and seven from King George Island-Keller Peninsula (with 111 leaf fragments investigated)—Antarctica. The fungal isolates used in this study were deposited in the Bruno Edgar Irgang Herbarium (HBEI—https://sites.unipampa.edu.br/brherbariohbei/ (accessed on 21 March 2018)) of the Universidade Federal do Pampa-São Gabriel (UNIPAMPA, São Gabriel, Brazil).

### 2.3. Morphology

Fungal macroscopic parameters (colony color, texture, reverse color, border type) and colony diameters were observed in three different media. Colors follow the specification proposed by the OACC [23]. All isolates were inoculated in the following media: PDA, Sabouraud Agar (Merck^®^ KGaA, Darmstadt, Germany), and Grass Extract Dextrose Agar (GE). The Grass Extract Dextrose Agar was obtained by grinding 100 g of fresh leaves of Deschampsia antarctica into 400 mL of distilled water. Then the extract was filtered, 20 mL of its content was added to 100 mL of culture medium, with 10 g of dextrose and 12 g of agar added (quantities for one liter). The final volume was sterilized at 121 °C for 20 min and poured into sterile Petri dishes [24]. All media were incubated at 5, 10, 20, and 23.5 ± 1 °C. The morphological and microscopic characteristics were evaluated from 15–30 days. Media under the same conditions were used to determine the microscopic characters (hyphae, conidiophores, chlamydospores and conidia), and measures were obtained by determining the length/width of individual characters. All analyses are made under slides in light microscopy under 100× oil immersion objective and imaging was performed on a Zeiss Axio Imager A2 (Carl Zeiss, Oberkochen, Germany) equipped with Axiocam MRc system (Carl Zeiss, Oberkochen, Germany). 

### 2.4. Molecular Analysis

Genomic DNA was extracted using the DNeasy Plant Mini Kit (Qiagen, Hilden, Germany), according to the manufacturer’s instructions. The internal transcribed spacer (ITS) region was amplified with universal primers for ITS1 (5′-TCCGTAGGTGAACCTGCGG-3′) and ITS4 (5′-TCCTCCGCTTATTGATATGC-3′) [25]. For ribosomal large subunit (nLSU) analysis, rDNA primers for NL1 (5′-GCATATCAATAAGCGGAGGAAAAG-3′) and NL4 (5′-GGTCCGTGTTTCAAGACGG-3′) were used [25,26]. The PCR procedure for ITS was as follows: initial denaturation at 95 °C for 3 min, followed by 35 cycles at 94 °C for 40 s, 54 °C for 45 s, and 72 °C for 1 min, followed by a final extension at 72 °C for 10 min. In addition, the PCR procedure for nLSU was as follows: initial denaturation at 95 °C for 5 min, followed by 30 cycles at 95 °C for 45 s, 57 °C for 45 s and 72 °C for 1 min, with a final extension at 72 °C for 10 min. The PCR products were purified using the Wizard^®^ Plus SV Miniprep DNA Purification System (Promega, Madison, WI, USA), and sequenced using a ABI-Prism 3500 Genetic Analyzer (Applied Biosystems, Waltham, MA, USA) with the same primers. The sequences obtained were adjusted using Bioedit software v. 7.0.5.3 [27], and a consensus sequence was obtained using Lasergene SeqMan software (DNASTAR/Inc., Madison, WI, USA). Representative consensus sequences were deposited into GenBank under the accession numbers: nLSU—MF628023, MF628257, MF628108, MF629819 and ITS—MF629817, MF629818.

#### Molecular Identification Analysis

To identify species by rDNA sequencing based on ITS and nLSU, the consensus sequences were aligned with sequences from related species retrieved from the NCBI GenBank database using BLAST [28]. The closest matched sequences with query cover and maximum identity ≥ 96% and ≥90% for ITS and LSU sequences, respectively, with an e-value ≥ 0, were included in the phylogenetic analyses. The dataset was used as the outgroup *Preussia minima* (Auersw.) Arx for ITS (MW090811.1) and nLSU (AY510392.1). Sequences were aligned with ClustalW as implemented in MEGA v. 6.06 [29]. Prior to phylogenetic analyses, ambiguous sequences at the start and end were trimmed to optimize the alignment. Bayesian inference (BI) was employed to perform phylogenetic analyses of the two aligned datasets. Bayesian analyses were conducted on the aligned data set using BEAST v. 1.8.3 software [30]. The Hasegawa-Kishino-Yano model of equal base frequencies was used for ITS and the Tamura-Nei model for nLSU dataset. In order to identify the posterior probability tree a 10 million Markov Chains Monte Carlo (MCMC) was run, and trees were sampled every 1000 generations. Tracer v1.6 [31] was used to evaluate the effective population size (ESS > 100), and TreeAnnotator v1.8.3 (from the BEAST package) [30] was used to condense the information from the trees sampled by MCMC. The fungal classification followed Onofri et al. [2], MycoBank (https://www.mycobank.org (accessed on 25 April 2018)) and the Index Fungorum (http://www.indexfungorum.org (accessed on 25 April 2018)).

### 2.5. Growth Experiments

Mycelia disks of 4 mm diameter for the four isolates studied were re-cultured in the same culture media from previous experiment. The plates were incubated at 5, 10, 20, and 23.5 ± 1 °C in the dark. Plates containing the mycelium for each of the species in each culture medium and temperature tested were performed in triplicate [32].

Radial mycelial growth was measured using a digital caliper from the back of the plate in four-line directions at 45° to each other in the 2 sectors (0, 45, 90, 135, 180, 225, 270, 315°) at 24-h intervals for each measurement. This was the first reading performed after the 4th day of incubation. The mean length was calculated for each treatment and isolated fungus obtained from the leaves of *D. antarctica*. The last growth measure was performed when the first isolate reached the border of one of the Petri dishes. This occurred thirteen days after the first length measurement. For the UNIPAMPA 006 isolate, it was not possible to perform statistical tests, since no growth was observed at 5, 10, and 23.5 °C, possibly due to the methodology used for measurement.

The experiment was conducted in a completely randomized manner. The data were analyzed by analysis of variance (ANOVA) [33], and the means were compared using the Tukey test (*p* < 0.05) of probability, assuming that the data are normal. Verification of the normality of the data was performed as proposed by Shapiro-Wilks [34,35,36]. When the data were not normal, they were transformed using the Tukey’s Ladder of Powers transformation method [37]. All statistical analyses were performed in the R computational environment (R Core Team 2017) with RStudio software [38].

## 3. Results

Four distinct fungi were isolated directly from the leaf fragments of four individual plants in Half Moon Island (246 leaves investigated), and seven from King George Island-Keller Peninsula (111 leaves investigated). The leaves of *D. antarctica* revealed four fungi morphospecies, as some thus isolated from both sample sites demonstrated the same morphology. These isolates were labelled as follows: UNIPAMPA 004, UNIPAMPA 005, UNIPAMPA 006 for Half Moon Island, and UNIPAMPA 007 for King George Island.

### 3.1. Macro- and Micro-Morphological Analyses

After detection of the preliminary genetic and morphological characteristics, we identified the fungus UNIPAMPA 004 as belonging to the genus *Pyrenophora*, UNIPAMPA 005 to the genus *Phaeosphaeria*, UNIPAMPA 006 to the genus *Neoascochyta*, and UNIPAMPA 007 to the *Alternaria* sp. For the *Alternaria* isolate, we preferred not to continue with the other analyses, as it was impossible to identify the species of the genus to which our isolate belongs correctly. This species continues to be studied and will be the target of further research in development in our group. However, we still left this OTU in the phylogenetic trees to contribute to the positioning of the other species studied. The macro- and micro-morphological characteristics of the other isolated fungi were evaluated on three media as described below.

#### 3.1.1. *Pyrenophora* cf. *chaetomioides* (UNIPAMPA 004)

The colonies grew at all temperatures, were cottony, and had white (oac909) or orange (oac649) edges, with a grey centre [(oac906) to (oac761) or (oac739-oac746/oac764-oac765), reverse darker -oac761)] (Figure 1). Hyphae transformed into chlamydospores (Figure 2k). Chlamydospores were terminal at 10.5–22.2 × 3.8–10.9 μm in size or catenulate and larger (15–29 × 9–24.5 μm in size) at 10 °C (Figure 2b–i). The pigment was evident at higher temperatures and dissolved in 5% potassium hydroxide (KOH) (Figure 2a,j).

Examined material: ANTARCTICA, South Shetland Archipelago, Half Moon Island, Austral Summer of 2014, A. B. Pereira (UNIPAMPA 004).

#### 3.1.2. *Septoriella elongata* (UNIPAMPA 005)

*Septoriella elongata* demonstrated colonies with borders presenting a hyaline margin (oac857) of up to 1 cm large that was plain and complete, with a white center (oac909) surrounded by yellow (oac853) or grey (oac908) mycelium. The reverse had colors from oac908 to oac763-oac637 or oac794-oac763 and all growing colonies are cottony (Figure 3). There were also hyphae hyaline to melleous of 4–11 µm in diameter, with sinuose walls in the terminal (Figure 4d). There were also conidiophores and conidia at terminal branches, with immature long ellipsoid to cylindrical conidia like those of *Stagonospora* sp. (Figure 4a–c). A net-like hyphae resembling a nematode capture loop was found alone at 10 °C in GE (Figure 4e).

Examined material: ANTARCTICA, South Shetland Archipelago, Half Moon Island, Austral Summer of 2014, A. B. Pereira (UNIPAMPA 005).

#### 3.1.3. *Neoascochyta paspali* (UNIPAMPA 006)

Growth was observed only at 20 °C in PDA and SAB culture media. Colonies generating cottony whitish tufts were formed through radially disposed hyphae, with gray and white (oac866/oac903) and reverse (oac908). There were also hyphae of 2–8 μm in diameter. Tufts were also identified in the mycelium in PDA (Figure 5a). The oldest mycelium had a higher proportion of pigmented hyphae (Figure 5d) and rare chlamydospores, which were 6–15 × 5.5–8 μm in size (Figure 5b,c).

Examined material: ANTARCTICA, South Shetland Archipelago, Half Moon Island, Austral Summer of 2014, A. B. Pereira (UNIPAMPA 006).

#### 3.1.4. *Alternaria* sp. (UNIPAMPA 007)

*Alternaria* sp. demonstrated colonies that were white to slightly pink in color (oac550) or beige (oac859-oac866), but the reverse was densely pigmented from oac756-oac796 to oac792. They were also plane to undulate and cottony (Figure 6). Primordia of conidiophores and conidia were formed at 5 °C in SAB culture medium, with intercalary chlamydospores at 8–8.5 × 4–5 μm at 5 °C in SAB (Figure 7a–c). Microsclerotium (Figure 7f) developed in PDA. There were also several round to ellipsoid perithecia formed only at 5 and 10 °C in PDA (Figure 7d). They had hyaline transversely multisepted ascospores that were 28–42.5 × 16.5–20.5 μm, with strangled septa that were immature (Figure 7e,i), with asci at 10 °C in PDA (Figure 7j). The GE culture medium facilitated the development of chlamydospores at all temperatures with 9.8–25 × 9–14 μm (Figure 7k–n). Similarly, this proportioned the development of immature conidia with two to three transverse septa that were 24 × 12 μm in size at 20 °C in GE (Figure 7o). Chlamydospores eventually formed dense clusters, probably microsclerotia or perithecia at 23.5 °C in GE (Figure 7p), and few conidia with transverse and one oblique septa with 31.5 × 7.8 μm (Figure 7q,r) were also observed at 23.5 °C in GE.

Examined material: ANTARCTICA, South Shetland Archipelago, Half Moon Island, Austral Summer of 2014, A. B. Pereira (UNIPAMPA 007).

### 3.2. Phylogenetic Analysis

To clarify the taxonomic position of the species, we performed a phylogenetic study based on the sequences of the ITS and nLSU regions. The sequences obtained from fungal cultures resulted in BLASTn hits for endophytic and pathogenic fungi. The isolates were considered as belonging to the *Pyrenophora*, *Phaeosphaeria*, *Neoascochyta* and *Alternaria* genus after a comparison of their nucleotide sequences revealed an identity above 90% for ITS regions of rDNA and nLSU.

Detailed phylogenetic analysis of the ITS region of the UNIPAMPA 006 and UNIPAMPA 005 sequences with the nearest taxa obtained from GenBank showed that UNIPAMPA 006 forms a distinct cluster close to species *Neoascochyta paspali* (NR135970) and *Phoma paspali* (KT309957), this last one being a basionym for *N. paspali*. In addition, the ITS sequence of UNIPAMPA 005 was grouped with the *Septoriella elongata* (KM491546 as *Phaeospheria elongata*) species.

The fungal isolates of UNIPAMPA 004 and UNIPAMPA 007 were not included in the ITS phylogenetic analysis because the sequence presented low quality and a smaller size than another homologous find in Genbank. Moreover, in the initial Blast survey, these sequences showed an e-value > 0 in the BLASTn query, which could have generated conflicts during the alignment with other sequences of fungi.

The total number of sequences of the ITS rDNA region compared to sequences associated with Antarctic grass leaf fungi was 31. Based on our results, the Bayesian Inference (Figure 8) tree in this dataset with two distinct clusters is supported (A and B). Cluster A (Figure 8) is comprised of 16 endophytic/pathogenic fungi sequences. Within this cluster were grouped sequences of fungi principally reported as endophytic. Of these taxa, all are identified as belonging to the *Phaeosphaeria* genus. These species were close to the UNIPAMPA 005 isolate. *Septoriella elongata* (KM491546) was collected from dead wood in Italy [39,40]. The analysis of this clade is supported by a posterior probability of 0.99, indicating that our species is *Septoriella elongata*.

The species that comprised cluster B (Figure 8) included 14 species of fungi that corresponded to endophytes and pathogens. These species were close to the UNIPAMPA 006 isolate and the phytopathogenic fungi *Neoascochyta paspali* (NR135970). In addition, most of the taxa are continuously present in the environment as saprobic soil organisms [41], *Neoascochyta europaea* (KT389510), and *Neoascochyta graminicola* (KT389518) fungus associated with plants and soil [42]. This clade was heavily supported (PP = 0.99).

Isolated UNIPAMPA 006 was inferred together with a fungus identified as *Neoascochyta paspali*. The genus *Neoascochyta* is ubiquitous and species-rich, with species occurring on a diverse range of substrates, including soil, air, plants, animals, and humans. The posterior probability supports that the isolated belongs to the *Neoacochyta paspali* species complex.

A total of 47 sequences from the LSU region were compared to the sequences obtained in this study (UNIPAMPA 004, UNIPAMPA 005, UNIPAMPA 006 and UNIPAMPA 007). The sequences were the result of the BLASTn search. The trees generated by BI analysis based on the nLSU dataset were similar in topology with the ITS region. The phylogenetic tree inferred clearly showed the formation of four large clusters (A, B, C and D) (Figure 9).

The first cluster (Figure 8A) included 11 species that are mostly comprised of pathogens. Our samples groups with sequences close to *Septoriella elongata* (KM491548) [43]. This clade was heavily supported (PP = 0.99) (Figure 9). These data corroborate the analysis carried out for the ITS region. Cluster B comprises 11 species that corresponded to pathogens and endophytes of Poaceae. The isolated UNIPAMPA 006 was grouped with sequences of the *Neoascochyta paspali* (GU238124), which belongs to relevant phytopathogenic fungi, including a series of pathogens with quarantine status [44] (Figure 8). Although most taxa are continuously present in the environment as saprobic soil organisms, many species switch to a pathogenic lifestyle when a suitable host is encountered [41]. This clade was heavily supported (PP = 0.99). These data corroborate the analysis carried out for the ITS region. Another 13 species were grouped at cluster C. These species were close to the UNIPAMPA 007 isolate. The sequence of the isolated fungus is related to *Alternaria chlamydospora* (KC584264) which is known as a severe plant pathogen that cause significant losses on a wide range of crops [45] and *Alternaria oregonensis* (KC584292). This clade was hardly supported (PP = 0.48). The genetic distances obtained do not allow us to ascribe our samples to any of the analyzed species of *Alternaria* mentioned above, suggesting that further analysis of a new species is necessary to the establish a taxonomically valid description for this isolate. The last cluster (Figure 9, group D) included 12 species of fungi classified as endophytes and pathogens (Figure 9). Our isolate grouped closer to *Pyrenophora chaetomioides* (JN940075) which was isolated mainly from the Poaceae species. This finding has posterior probability support (0.63). It is insufficient to consider our sample as belonging to this species, so we treated it as *Pyrenophora* cf. *chaetomioides* because we do not have sufficient morphological and molecular evidence for decisive identification. For this reason of the indefinite taxa, no result or discussion of this isolate is presented here. Complementary molecular analyses are being performed, such as whole genomic sequencing, and marks on the effect of temperature on this isolate will be presented in due course when the authors are confident of the taxonomic determination of this isolate.

### 3.3. Effect of Temperature on Growth

Of the four isolates selected for the present study, only *Neoascochyta paspali* did not respond to in vitro growth assays. The samples for this isolate did not develop enough biomass for the measurements, so we decided to exclude it from the analyzes so as not to harm the rest of the tests. For the other isolates, the results are described below:

#### 3.3.1. *Pyrenophora* cf. *chaetomioides*

The mean radial growth of *Pyrenophora* cf. *chaetomioides* in SAB medium at 5 °C was the lowest among the three media tested (mean = 0.165 mm). The other two media showed a higher growth in this same temperature, with averages of 0.905 mm and 0.537 mm for PDA and GE, respectively. The statistical test indicated a significant difference between these averages at this temperature. Furthermore, GE was identified as the better culture medium for the faster growing of this isolate at 20 °C, since it showed an average growth of 2.361 mm. At 23.5 °C, the fungus presented a lower growth in the three media (mean PDA = 0.419 mm, mean SAB = 0.281 mm, and mean GE = 0.703 mm), indicating that this species is more sensitive at this temperature (Figure 10). The isolate with the selected culture medium (GE), at all temperatures tested, demonstrated significant differences between temperatures (Table 1). The temperature with the highest mycelial growth was 20 °C; considering this criterion, its mean differed statistically from other temperatures. However, the other three temperatures showed no significant differences (Table 1) in their average growth for the GE medium.

#### 3.3.2. *Septoriella elongata*

In the in vitro experiment, at a temperature of 23.5 °C, no growth was identified for the *Septoriella elongata* isolate. No significant differences were detected in the average growth of the colonies of the isolate (Figure 11) between the different culture media at temperatures 5 and 10 °C. The isolate at 20 °C showed significant differences between the media used, and the PDA medium demonstrated the highest growth. Samples of the isolate in PDA medium showed the highest mycelial growth at 10 °C, but statistically there were no significant differences between the lowest temperatures tested in the present study. The analysis of the ideal culture medium (PDA) for this species between different temperatures resulted in a significant difference (Table 1). The temperature for the selected medium with the highest growth was 10 °C (mean = 2.295 mm), followed by 20 °C (mean = 1.642 mm), and the lowest growth was observed at 5 °C (mean = 1.111 mm).

#### 3.3.3. *Alternaria* sp.

After 13 days of evaluation of the radial mycelial growth, and from the analysis of the Tukey test and ANOVA at a 5% confidence level, it was observed that the mycelial growth did not differ when this isolate was kept at 5 °C in all culture media studied (PDA, SAB, and GE). However, for the other temperatures (10, 20 and 23.5 °C), significant growth differences were observed between the three culture media (Figure 12). At 20 °C, the isolates of *Alternaria* sp. presented the highest growth (mean = 1432 mm), followed by a temperature of 23.5 °C (mean = 1.325 mm) and 10 °C (mean = 0.823 mm), all in the middle of the PDA culture (Figure 12). Therefore, this medium was considered the best for determining the ideal growth temperature of this fungus isolate. At lower temperatures, the isolate maintained its growth with the formation of reproductive structures, indicating that it is a psychrophilic fungus (Figure 7a–j). From the individual analysis of the PDA culture medium, considering the four temperatures studied, there were no significant differences between temperatures of 20 and 23.5 °C (Table 1).

## 4. Discussion

The major group of fungal endophytes in plants is represented by species of Ascomycota, which was confirmed in this study. The genera found are also widely distributed [46].

Previous studies have revealed a diversity of endophytic fungal communities associated with plants living in tropical, temperate, and boreal ecosystems, and their frequency seems to decrease in cold regions [47]. Saikkonenet al. [48] demonstrated a low incidence of endophytes from *Deschampsia flexuosa* (L.) Trin. and *Deschampsia cespitosa* (L.) P. Beauv. in the cold regions of Finland. Rosa et al. [14] isolated 18 fungi as endophytes from 273 leaf fragments of the Antarctic hairgrass resulting in 18 species. Our results also point to a small diversity of fungi associated with the *D. antarctica* leaf (four). However, our sampling effort was limited to two single islands in the Maritime Antarctic. This may reflect low isolated diversity since endophytic fungi can be restricted by geography but not by host [49]. Another possibility is that species diversity may vary with environmental factors at sample sites, but further investigation is required to confirm this.

One of the fungal genera reported in *D. antarctica* leaves is *Phaeosphaeria*, which is known as a pathogen that causes leaf spots on grasses and some other monocots. Dennis [50] was the first to report species of this genus from areas near Antarctica (South Georgia –sub-Antarctic Island). Some species are very specialized while others have a large host spectrum [51]. *Phaeosphaeria* is distributed over all South, Central, and North America as well as Africa and Asia [52,53]. The most related species to the *Phaeosphaeria* isolate identified in this study are *Phaeospharia elongata* (*Septoriella elongata*), associated with terrestrial or near freshwater grasses [43]. Some species from this genus were replaced with other genera recently [54], for this reason we prefer to use *Septoriella elongata* in the present study. Putzke and Pereira [4] described *Phaeosphaeria deschampsii* in Antarctica as a new species, showing that this genus is also formed by several unknown species in the area and associated with Antarctic hair-grass.

*Phaeosphaeria/Septoriella* are parasites found in many grass cultures. The species are usually very specialized and can cause deadly diseases. Some species have a wide range of hosts, mostly among Poaceae and other monocots (Cyperaceae, Juncaceae, etc.), as well as *Lycopodium* and *Equisetum* [55]. The anamorph often belongs to species of the genus *Stagonospora*. These fungi normally grow in leaves or floral parts of Poaceae [56]. Our isolate *Septoriella elongata* is an anamorph of *Stagonospora* characterized by its solitary and hyaline cylindric conidia, and plane margins in PDA. The aerial mycelia are scarce, with a cream color at the beginning that turns pallid to olivaceous gray and then whitish with a dark reverse [57,58], such as the UNIPAMPA 005 isolate.

The genus *Phaeosphaeria* is known to present pathogenic and endophytic plant species. In addition, this genus presents a generalized distribution in grain crop areas [59,60]. According to Jankowiaket al. [61], species belonging to the genus *Phaeosphaeria* were isolated from root fragments and cotyledons of *Abies alba* and incubated at temperatures of 22–25 °C. Cervelattiet et al. [62] reported that the optimum temperature for *Phaeosphaeria* maydis ranges from 12 to 22 °C. The UNIPAMPA 005 isolate presented characteristics similar to those observed in the previous study, growing at temperatures of 5, 10, and 20 °C, with the highest growth in 10 °C of all the media used (BDA, SAB and GE).

The anamorphic genus *Phoma* includes many important pathogenic fungi [59]. Aveskamp et al. [45] isolated *Phoma paspali* (*Neoascochyta paspali*) from the *Paspalum notatum grass*. Approximately 50% of *Phoma* species, redescribed by Boerema [63], were recognized as relevant phytopathogens. The morphologic characteristics of this fungus in PDA include regular margins with hyaline and white mycelia and colonies presenting hyaline to white radial spherical tufts that were densely clustered at the top, and later changed color to gray [64]. These results agree with our study. Unicellular dark brown to olivaceous terminal chlamydospores in aerial erect hyphae were described in Boerema et al. [63], which also correlates with the isolated UNIPAMPA 006 identified in this study. Phylogenetic and morphological analyses demonstrated that our isolate was *Neoascochyta paspali*, with a posterior probability of 0.99.

The genus *Neoascochyta* is one of the largest fungal genera, with more than 3000 species described. Species belonging to the genus *Neoascochyta* are often encountered as plant pathogens (mostly causing leaf spots) and as endophytes that utilize various hosts (including corn, citrus, and sorghum) [65]. The species most related to the UNIPAMPA 006 isolate is *Neoascochyta paspali* (=Phoma paspali). This species has not previously been reported in Antarctica and is considered an indigenous pathogen of grasses in Australia, New Zealand, and Europe [41]. Zhang and Yao [66] detected 31 known fungal species, most of which were originally reported in other habitats as endophytes in the leaves and stems of Arctic plants. *Phoma herbarum*, for example, was a widespread saprotroph and pathogen of plants, and has been found in diverse environments including Antarctica [67]. These results indicate the presence of specific psychrophilic and psychrotrophic fungi in various habitats of cold ecosystems. Furthermore, the wide distribution of these fungi suggests that they may be capable of long-distance dispersal.

*Pyrenophora* ssp. are another plant pathogen described as graminicolous, causing leaf spots in agronomically-important plants [68]. *Pyrenophora antarctica* was detected on Kerguelen Island (sub-Antarctic) on *Festuca antarctica* grass [60]. The UNIPAMPA 004 isolate is related to *Pyrenophora chaetomioides*, a specialized pathogen infecting various species of oats (*Avena* spp.) and some grasses [69]. Onofri et al. [2] reported no species of this genus to Antarctica as this was the first reference to the area.

The genus *Pyrenophora* is responsible for helminthosporiose leaf blight in wheat and barley, which causes a disease with great economic importance. These fungi can survive as mycelium in seed endosperms including during water stress, thus colonizing the radicular system since it is activated during seed germination [70]. As described by Farias et al. [67] and Benslimane et al. [71], this genus presents a significant range in conidia dimensions, including being formed directly from chlamydospores. The UNIPAMPA 004 isolated presented only rectangular to globose chlamydospores, terminal or intercalary, which makes microscopical identification impossible since no conidia was observed.

According to Ruisi et al. [17], geographic isolation, combined with environmental stress, make Antarctica an ideal location to research new species of endemic fungi. Endophytic fungi in relatively extreme environments as well as phylogenetically distinct plant strains are promising sources for discovering undescribed species, which is important in understanding fungal diversity [47].

The use of macro- and microscopic characters of anamorphic cultures usually does not offer enough information for taxonomic identification [72,73]. The UNIPAMPA 007 isolate exhibits morphological characteristics very close to those in anamorphic stage. The morphology that agrees with the teleomorph are the hyaline spores of the genus *Pleospora* [74,75]. This isolate presented characteristics close to those described by Grum-Grzhimaylo et al. [76], such as the development of narrow conidia and terminal or intercalary chlamydospores.

Studies using plant extracts as culture medium have been carried out with the objective of verifying the development of morphological structures. The genus *Pyrenophora* is mainly characterized as plant pathogens, particularly of Poaceae. Borba et al. [24] demonstrated that the temperature for a better mycelial development for this genus in culture medium supplemented with grass extract is around 25 ± 1 °C. Our study showed that in the GE culture medium UNIPAMPA 004 grew better, and shows the highest mycelial growth in temperatures of 20 °C. Linhares et al. [77] demonstrated that 22 °C was the best incubation temperature for pathogens of the genus *Pyrenophora*. Khouri et al. [78] evaluated the effect of grass extracts in Ascomycota growth and concluded that *Cynodon dactylon* (Poaceae) and *Digitaria decumbens* (Poaceae) grass promoted better fungal growth. GE was supplemented with *D. antarctica* leaves, demonstrating greater mycelial growth at temperatures of 20 and 23.5 °C compared to other media. In addition, preference for plant species may be related to the nutritional requirement of the fungus [77,78]. According to Reis [79] species of this genus can be inoculated in PDA culture medium or supplemented with plant extracts, as these can provide carbon and sugar for their development [80].

The results of growth tests at different temperatures suggest that the fungi associated with *Deschampsia antarctica* in the Half Moon and King George Islands can grow at temperatures of 10 and 20 °C. Tosi et al. [81] demonstrated that most of the fungi isolated from mosses in Victoria Land could grow at temperatures ≤5 °C but exhibited optimum growth between 10 and 24 °C. Most endophytic fungi isolated from Antarctic mosses are also psychrotrophic and psychrophilic [32]. In addition, fungi may exhibit morphological adaptations, an example being the predominance of non-sporogenous mycelia at low temperatures. These physiological and morphological mechanisms were reported for fungi present in Antarctica as well in other environments [17].

Based on the observations of Newsham et al. [82], future warming in Antarctica will lead to increases in fungal populations, and this could have negative consequences on plant productivity, in the case of these endophytic fungi spread in warmer environments finding non-responsive plants. These data agree with our study, considering that the *Pyrenophora* cf. *chaetomioides* isolate showed higher mycelial growth at 20 °C, and *Septoriella elongata* at 10 °C.

## 5. Conclusions

The Antarctic continent has unique environmental conditions that allow the isolation and identification of endemic and new species of fungi. Applying molecular and morphological approaches to the fungi isolated from *Deschampsia antarctica* we identified endophytic/pathogenic fungi *Septoriella elongata*, *Pyrenophora* cf. *chaetomioides Alternaria sp.* and *Neoascochyta paspali* relating those species to cold environment and classifying them as psychrophilic organisms. The study of such a group of species is very interesting since they could elucidate issues related to environmental changes and those associated with communities of antarctic plants.

## Figures and Tables

**Figure 1 life-12-01501-f001:**
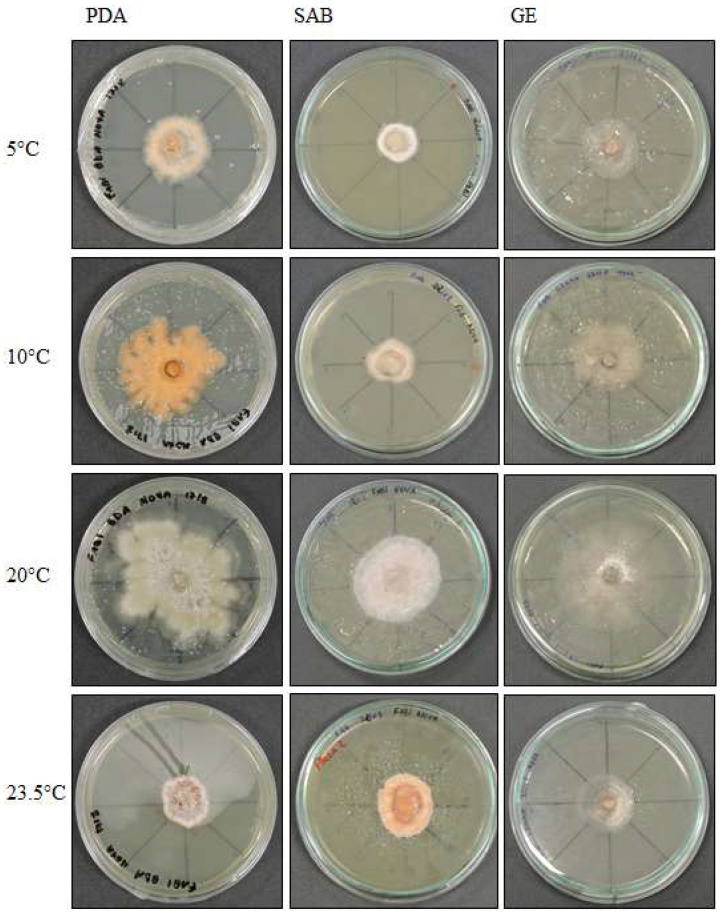
*Pyrenophora* cf. *chaetomioides* colony morphology on three media (Potato Dextrose Agar—PDA, Sabouraud Agar—SAB, Grass Extract Dextrose Agar—GE) and growth at different temperatures (5 °C, 10 °C, 20 °C and 23.5 °C).

**Figure 2 life-12-01501-f002:**
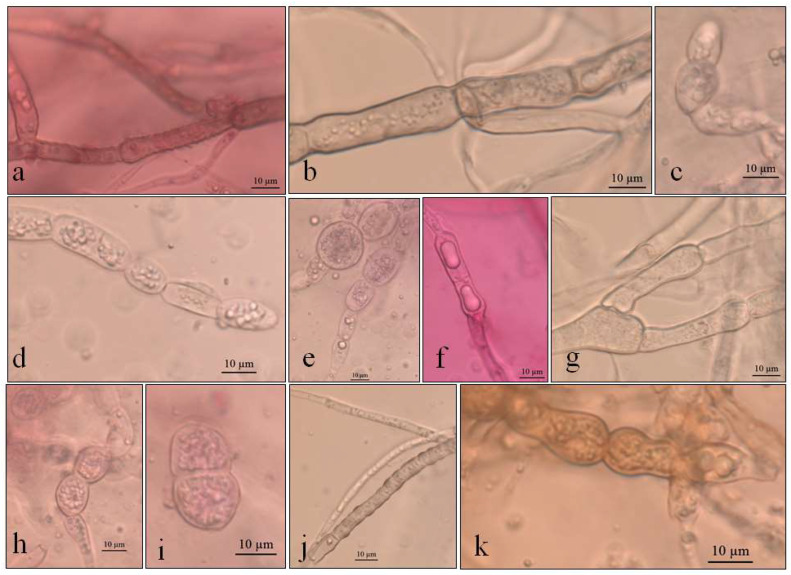
Optical micrographs images of *Pyrenophora* cf. *chaetomioides* (**a**) Hyphae pigmented (under in KOH preparation); (**b**) Chlamydospores main shape at 5 °C in PDA; (**c**) Chlamydospores terminal at 5 °C in SAB; (**d**) Chlamydospores catenulate and larger at 10 °C in SAB; (**e**) Chlamydospores at 10 °C in GE; (**f**) Chlamydospores catenulate at the terminal hyphae at 20 °C in GE; (**g**) Branched terminal hyphae found close to chlamydospores. (**h**,**i**) Globose chlamydospores at 23.5 °C in SAB; (**j**) Pigment in hyphae in PDA; (**k**) Hyphae turning into chlamydospores at 23.5 °C in SAB.

**Figure 3 life-12-01501-f003:**
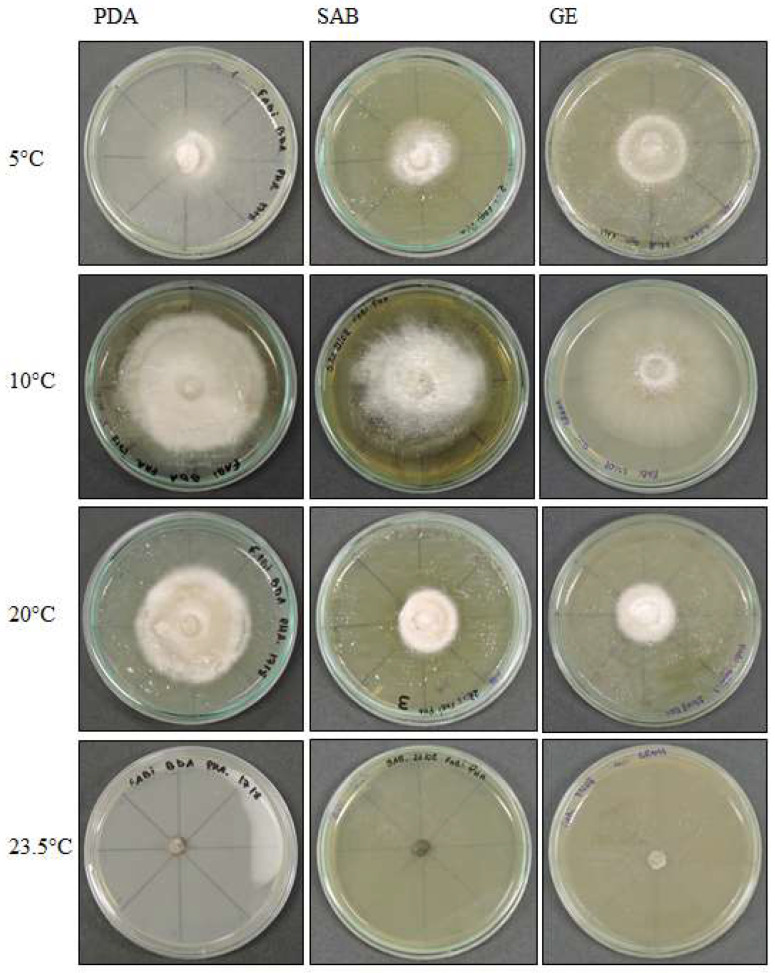
*Septoriella elongata* colony morphology on three media (Potato Dextrose Agar-PDA, Sabouraud Agar-SAB, Grass Extract Dextrose Agar-GE) and growth at diferent temperatures (5 °C, 10 °C, 20 °C and 23.5 °C).

**Figure 4 life-12-01501-f004:**
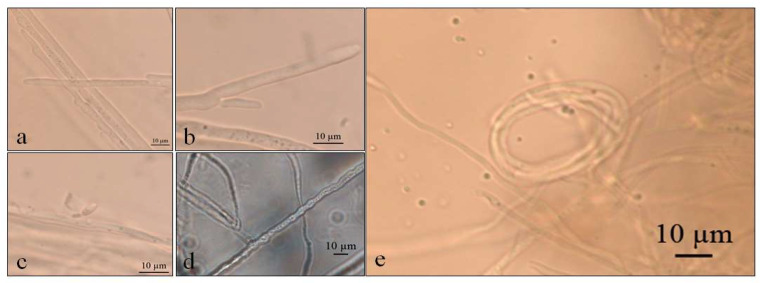
Optical micrographs of *Septoriella elongata* (**a**–**c**) Conidial development in culture medium GE; (**d**) Sinuose hyphae at 5 °C in PDA; (**e**) Net-like structure similar to nematode capture hook.

**Figure 5 life-12-01501-f005:**
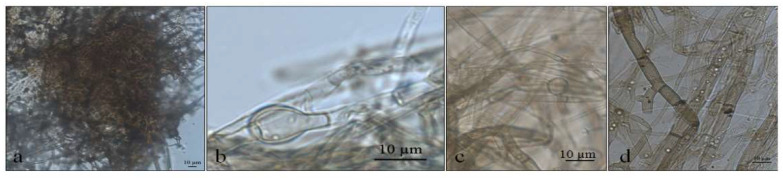
Optical micrographs of *Neoascochyta paspali* (**a**) tufts in the oldest mycelium in PDA; (**b**,**c**) Chlamydospores; (**d**) Pigmented hyphae.

**Figure 6 life-12-01501-f006:**
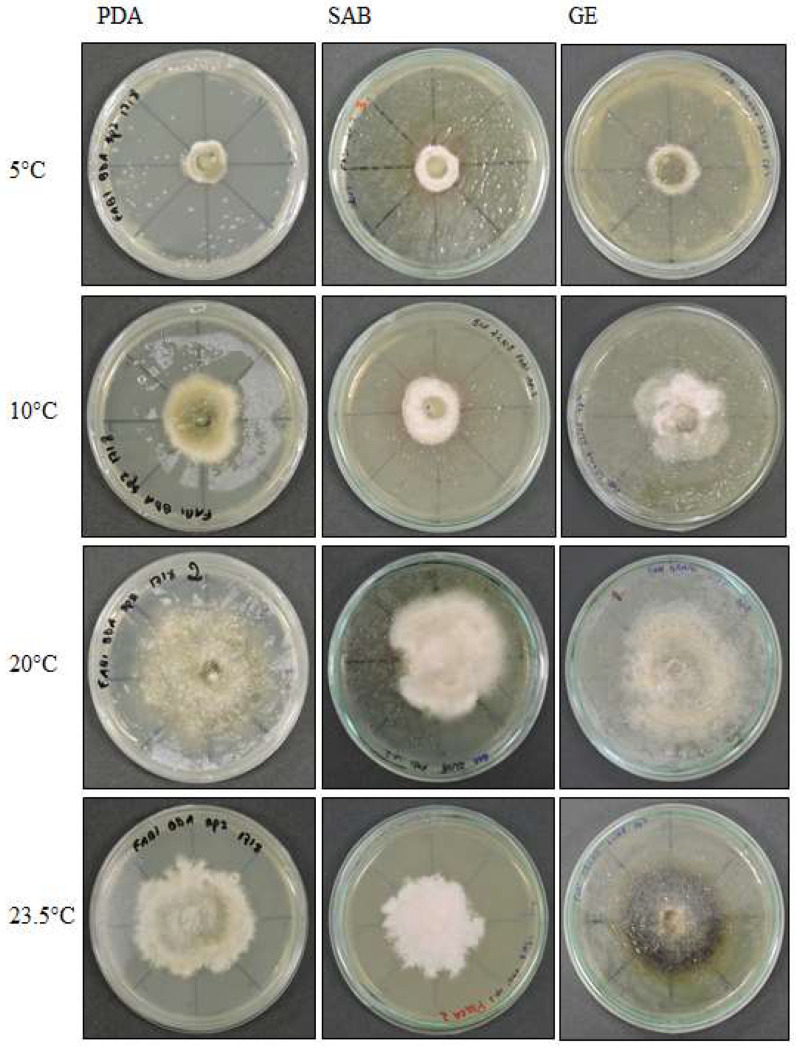
*Alternaria* sp. colony morphology on three media (Potato Dextrose Agar—PDA, Sabouraud Agar—SAB, Grass Extract Dextrose Agar—GE) and growth at diferent temperatures (5 °C, 10 °C, 20 °C and 23.5 °C).

**Figure 7 life-12-01501-f007:**
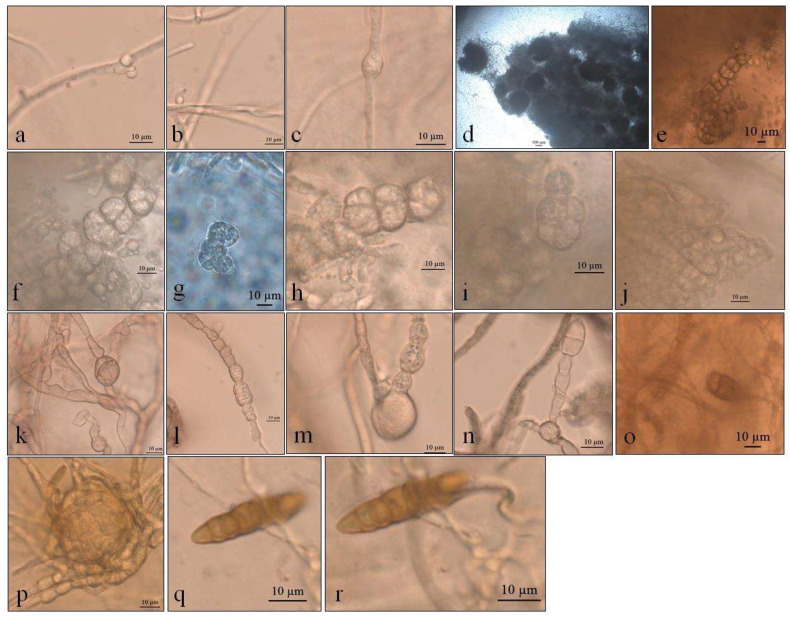
Optical micrographs of *Alternaria* sp. (**a**,**b**) Primordia of conidiophores and conidia at 5 °C in SAB; (**c**) Intercalarly chlamydospores at 5 °C in SAB; (**d**) Perithecia at 10 °C in PDA; (**e**–**i**) Ascospores in PDA; (**j**) Asci in PDA; (**k**–**n**) Gigant chamydospores in GE; (**o**) Immature conidia at 20 °C in GE; (**p**) Clamidospores forming microsclerotia or perithecia at 23.5 °C in GE; (**q**,**r**) Conidia at 23.5 °C in GE.

**Figure 8 life-12-01501-f008:**
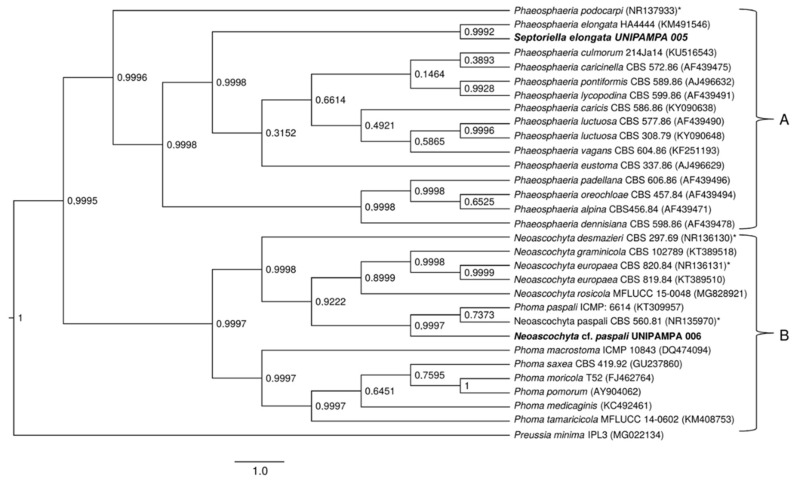
Phylogenetic tree showing the relationship among *Deschampsia antarctica* associated fungi and other fungal species. The tree was constructed based on the rDNA sequence (ITS1-5.8S-ITS2) fragment by using the Bayesian Evolutionary Analysis Sampling Trees. Scale bar reflect estimated number of 1.0 changes per site. The robustness of each node is represented by the posterior probability value obtained after 10,000,000 Monte Carlo Markov chains (MCMC). Sequences of type species (*). The tree was rooted using *Preussia minima* as outgroup due to being outside the clade of interest.

**Figure 9 life-12-01501-f009:**
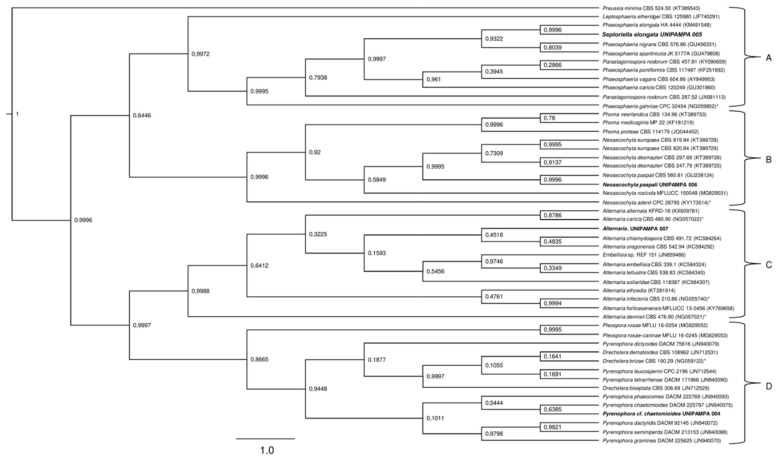
Phylogenetic tree showing the relationship among *Deschampsia antarctica*-associated fungi and other fungal species. The tree was constructed based on the nLSU region fragment by using the Bayesian Evolutionary Analysis Sampling Trees. Scale bar reflects the estimated number of 1.0 changes per site. The robustness of each node is represented by the posterior probability value obtained after 10,000,000 Monte Carlo Markov chains (MCMC). Sequences of type species (*). The tree was rooted using *Preussia minima* as outgroup due to being outside the clade of interest.

**Figure 10 life-12-01501-f010:**
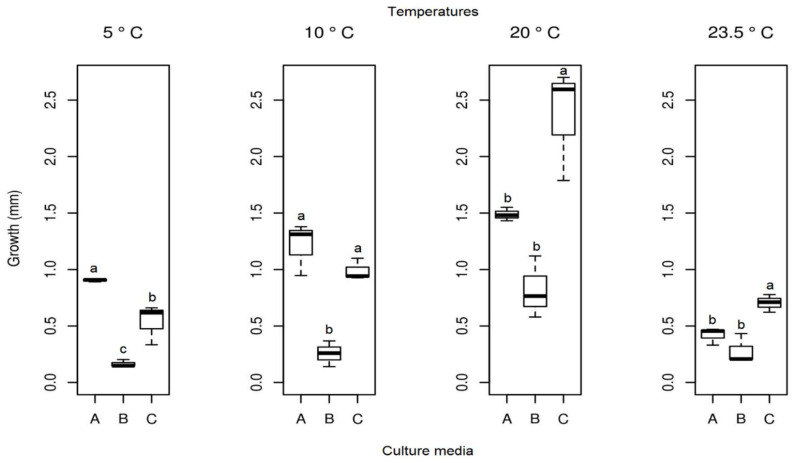
Boxplot of mycelial growth of the isolate *Pyrenophora* cf. *chaetomioides* for the four temperatures studied. Boxplots with the same letter (lowercase), at the same temperature, do not differ statistically from each other by the Tukey test at 5% confidence. A = PDA, B = SAB and C = GE.

**Figure 11 life-12-01501-f011:**
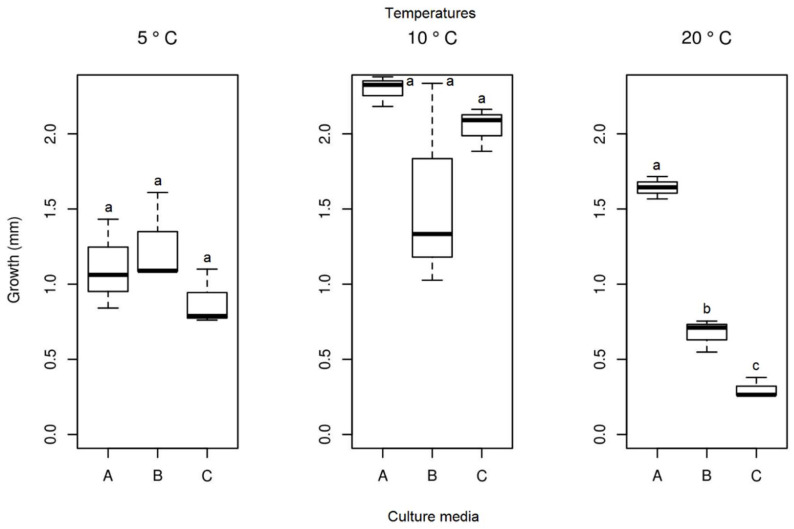
Boxplot of mycelial growth of the isolate *Septoriella elongata* for the four temperatures studied. Boxplots with the same letter (lowercase), at the same temperature, do not differ statistically from each other by the Tukey test at 5% confidence. A = PDA, B = SAB and C = GE.

**Figure 12 life-12-01501-f012:**
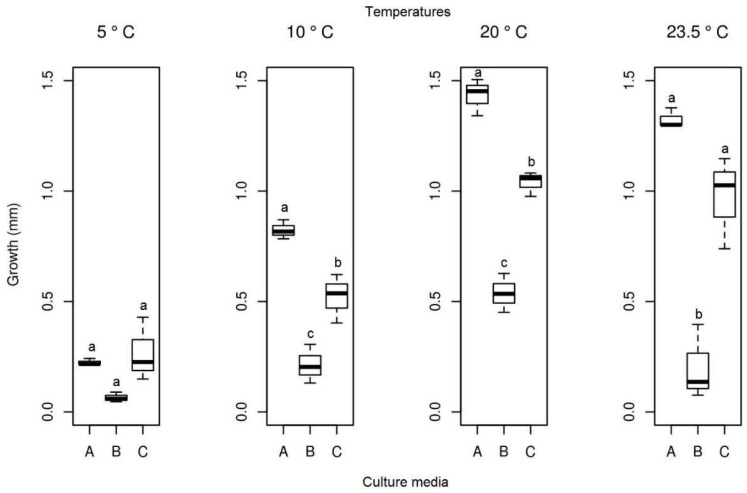
Boxplot of mycelial growth of the isolate *Alternaria* sp. for the four temperatures studied. Boxplots with the same letter (lowercase), at the same temperature, do not differ statistically from each other by the Tukey test at 5% confidence. A = PDA, B = SAB and C = GE.

**Table 1 life-12-01501-t001:** Radial mycelial growth for fungi isolated from *Deschampsia antarctica* leaves in PDA medium for *Alternaria* sp. and *Phaeosphaeria* sp, and in GE medium for *Pyrenophora* sp. *.

Treatments	*Septoriella elongata*	*Pyrenophora* cf. *chaetomioides*
5 °C	1.111 ^c^	0.537 ^c^
10 °C	2.295 ^a^	0.989 ^b^
20 °C	1.642 ^b^	2.361 ^a^
23.5 °C	-	0.704 ^bc^

* Different letters in the columns differ from each other values of significance levels in the Tukey test (α = 0.05).

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
