# Peer review of "Impact of the Temperature in Endophytic Ascomycota Isolated from Antarctic Hair-Grass"

_life, 2022, doi:10.3390/life12101501_

Round 1
Reviewer 1 Report
I have reviewed your ms. thoroughly, and found that the research can be accepted but only after major revision. Although your research is important, there are several major issues that should be corrected or improved: (1) The elaboration of all four isolated species should be balanced (there are major items missing for two of them); (2) You omitted the "big picture" in several parts of your ms. because of too narrow view (e.g. in the Abstract and Conclusion chapters, see comments in PDF); (3) Material and Methods chapter should be build on in several respects (see attached PDF): (4) The organisation of the text and contributed supporting items (figures, tables, diagrams etc.) must be improved; (5) Some microscopical elements are very likely erroneously interpreted. All details of the review (corrections, comments and suggestions) are highlighted in attached file below.

Author Response
POINT-BY-POINIT RESPONSE TO REVIEWER 1
RESPONSE: The authors made an extensive evaluation of the manuscript, seeking to follow the reviewer recommendations, as follow:
Reviewer recommendation (1) The elaboration of all four isolated species should be balanced (there are major items missing for two of them)
Response: Missing data on the results of morphological, phylogenetic, and growth response analyses for Alternaria were included. It should be noted that these were initially omitted by the authors' decision since the taxa was not identified until a specific epithet. After reviewing this reviewer's suggestions, we agreed that it was necessary to present these data for a broad understanding of the results obtained in the present study.
Reviewer recommendation (2) You omitted the "big picture" in several parts of your ms. because of too narrow view (e.g. in the Abstract and Conclusion chapters, see comments in PDF).
Response: Same answer above from item 1
Reviewer recommendation (3) Material and Methods chapter should be build on in several respects (see attached PDF):
Response: The authors evaluated and made some modifications to the methods. However, we believe that we have to be faithful in how the work was designed and developed, therefore, the way in which the methodology is presented seems to us to be the most adequate, in addition to the pertinent suggestions that were incorporated.
Reviewer recommendation (4) The organization of the text and contributed supporting items (figures, tables, diagrams etc.) must be improved.
Responses: Several parts of the ms were rewritten, follow the reviewer suggestions.
Reviewer reccomendation: (5) Some microscopical elements are very likely erroneously interpreted. All details of the review (corrections, comments and suggestions) are highlighted in attached file below.
Response: We re-evaluated and improved the manuscript accordingly.
Furthermore the manuscript were revised by two native speakers besides the sggestions made by the reviewer.
Reviewer 2 Report
These are important issues that require attention:
1. What are the results of the combined phylogenetic analyses of the ITS and LSU loci? Are they the same topology or are they different? If the authors could create a new phylogenetic tree using the combined ITS and LSU datasets and compare the topology to the individual ITS and LSU data, that would be excellent.
2. Phaeosphaeria elongata is not the official name, according to http://www.indexfungorum.org/. Change it to "Septoriella elongata", Phaeosphaeria elongata (Wehm.) Shoemaker & C.E. Babc., Can. J. Bot. 67(5): 1540 (1989).
PS: Species Fungorum current name:
Septoriella elongata (Wehm.) Y. Marín & Crous, in Marin-Felix, Hernández-Restrepo, Iturrieta-González, García, Gené, Groenewald, Cai, Chen, Quaedvlieg, Schumacher, Taylor, Ambers, Bonthond, Edwards, Krueger-Hadfield, Luangsa-ard, Morton & Moslemi 2019
3. In addition, I have quite some comments and corrections made on the PDF file, which is attached via orange or red highlighted.
4. In Figs. 7-8, it is too small and hard to see, particularly the relationship between fungi in this study and closely related taxa. It would be great if you could provide a bigger one.
5. Optionally, please provide a TreeBASE number based on the LSU or ITS tree (http://treebase.org/treebase-web/home.html). What is the number for the tree deposited in TreeBase?
6. The scientific names of all taxa need to be formatted in italics (check throughout the part of the reference
7. For general issues, the authors need to consult a native English speaker who is an academic one or send it to an English editing service before re-submission. Indeed, the manuscript needs to be re-corrected for grammar, phrasing, language, and punctuation and to maintain the author's meaning, structure, and logic as well as to optimize the writing style and flow.

Author Response
- What are the results of the combined phylogenetic analyses of the ITS and LSU loci? Are they the same topology or are they different? If the authors could create a new phylogenetic tree using the combined ITS and LSU datasets and compare the topology to the individual ITS and LSU data, that would be excellent.
Response to reviewer: Initially we thought of doing this. However, we found that Genbank (and other molecular databases) do not have the same regions for all species used in the comparisons, except for the outgroup. Thus, there is no way to build a consensus tree with the same topology in ITS and nLSU. Therefore, in order to include a greater number of species for the present comparison presented in the phylogenetic analyses, we prefer to use the trees separately.
- Phaeosphaeria elongatais not the official name, according to http://www.indexfungorum.org/. Change it to "Septoriella elongata", Phaeosphaeria elongata (Wehm.) Shoemaker & C.E. Babc., Can. J. Bot. 67(5): 1540 (1989).
PS: Species Fungorum current name:
Septoriella elongata (Wehm.) Y. Marín & Crous, in Marin-Felix, Hernández-Restrepo, Iturrieta-González, García, Gené, Groenewald, Cai, Chen, Quaedvlieg, Schumacher, Taylor, Ambers, Bonthond, Edwards, Krueger-Hadfield, Luangsa-ard, Morton & Moslemi 2019
Response: The names were changed in the ms.
- In addition, I have quite some comments and corrections madeon the PDF file, which is attached via orange or red highlighted.
Response: All suggestion were accepted and included in the new version. Thank you for your careful reading and we have done our best to respond to your suggestions.
- In Figs. 7-8, it is too small and hard to see, particularly the relationship between fungi in this study and closely related taxa. It would be great if you could provide a bigger one.
Response: The original figures are in 350dpi, but as the Life/MDPI model asks for the figures to be embedded, we believe that only in the final pdf will the images be more defined. If possible, we can provide the figure alone in a single pdf for your evaluation.
- Optionally, please provide a TreeBASE numberbased on the LSU or ITS tree (http://treebase.org/treebase-web/home.html). What is the number for the tree deposited in TreeBase?
Response: We tried to do it several times, but TREEBASE seems to be discontinued as it is not possible to make new insertions.
- Thescientific namesof all taxa need to be formatted in italics (check throughout the part of the reference
Response: All changed were made.
- For general issues, the authors need toconsult a native English speakerwho is an academic one or send it to an English editing service before re-submission. Indeed, the manuscript needs to be re-corrected for grammar, phrasing, language, and punctuation and to maintain the author's meaning, structure, and logic as well as to optimize the writing style and flow.
Response: The final ms were revised by a native spearker and we hope everything is ok now.

Round 2
Reviewer 1 Report
Please check the latin binominal in lines 499, 621, and 626. You have changed Phaeosphaeria elongata into Septoriella elongata (which was an excellent thing to do, since this is the correct name) except in line 488!
Author Response
All typo error has been corrected.

Reviewer 2 Report
The author has already addressed all of the issues raised in the comment. Please be patient while the journal takes the next step. Thank you for your patience throughout the review process.
Author Response
I just have to thank the reviewer's comments. In addition to qualifying the work, it helped us to see some problems that went unnoticed. Thank you very much.